# The Interaction between Antioxidants Content and Allergenic Potency of Different Raspberry Cultivars

**DOI:** 10.3390/antiox9030256

**Published:** 2020-03-20

**Authors:** Ewelina Hallmann, Alicja Ponder, Mateusz Aninowski, Tuya Narangerel, Joanna Leszczyńska

**Affiliations:** 1Institute of Human Nutrition Sciences, Department of Functional and Organic Food, Warsaw University of Life Sciences, Nowoursynowska 159c, 02-776 Warsaw, Poland; alicja_ponder@sggw.pl; 2Institute of Institute of Natural Resources and Cosmetics, Faculty of Biotechnology and Food Sciences, Lodz University of Technology, Stefanowskiego 4/10, 90-924 Lodz, Poland; mateusz.aninowski@edu.p.lodz.pl (M.A.); tuya.narangerel@dokt.p.lodz.pl (T.N.); joanna.leszczynska@p.lodz.pl (J.L.)

**Keywords:** allergenic potency, anthocyanins, flavonoids, organic, raspberry

## Abstract

Food allergies are a very serious problem among consumers. The most common food allergies involve animal products, but they can also involve fruits such as berries. We aimed to determine whether organic farming is useful for the production of high-quality and safe fruits. Three varieties of raspberries ‘Laszka’, ‘Glen Ample’ and ‘Polka’ from organic and conventional production (neighboring farms) were collected over the two years of the experiment. Quantitative and qualitative analysis of phenolic compounds was carried out, and the content of Bet v1 and profilin was determined. The organic raspberries contained a lower level of phenolic compounds, especially anthocyanins. Conventional fruits were characterized by a higher allergenic potency than organic ones. We found a strong link between their anthocyanin content and the allergy status of conventional raspberry fruits. Therefore, organically produced raspberries are safer for consumers.

## 1. Introduction

Raspberries are one of the most popular fruits in Europe. They are recognized by consumers as tasty and healthy fruits. Many studies have shown that regular fruit consumption can diminish the risk of many chronic diseases—such as neurodegenerative conditions, type 2 diabetes, some kinds of cancer, cardiovascular diseases, hypertension, overweight and obesity [1,2,3,4,5]—due to their biologically active compounds content. Plants in organic agriculture are cultivated without the use of artificial pesticides and mineral fertilizers. Only natural methods of plant protection and fertilization are allowed (Council Regulation (EC), 2007) [6]. Polyphenols are a large group of secondary metabolites produced by plants as a response to biotic and abiotic environmental stresses [7]. Organic raspberries contain significantly more bioactive compounds than conventionally raised berries [8]. The consumption of organic raspberries could, therefore, be more health promoting, due to the higher content of biologically active compounds in the fruits. On the other hand, it should be remembered that the consumption of raspberries carries the risk of food allergies. Reports on the allergenicity of small fruits (berries)—such as strawberries, raspberries, blackberries and blueberries—are still scarce, but they do exist [9]. During the last ten years, the prevalence of food allergies has increased up to 4% among adults and 6% among children. In berry fruits, a substantial proportion of the allergens have defense-related functions, and their expression is highly influenced by exposure to biotic and abiotic stress and diseases. Pathogenesis-related proteins (PRs) account for approximately 25% of plant food allergens, and some are responsible for extensive cross-reactions between plant-derived foods, pollen and latex allergens [10]. Bet v 1 is the most frequent cause of pollen-related food allergies [11], which are the most frequent type of food allergy in adults. The clinical symptoms observed are elicited by Bet v 1-induced IgE, which can then cross-react with a number of Bet v 1-related proteins from plant-derived foods. In raspberry fruits, only two allergenic proteins, including different isoforms, have been identified and described. The Rub i 1 and Rub i 3 allergens in raspberry (*Rubus idaeus* L.) belong to the PR-10 group (17 kDa) and are homologous to the major birch pollen Bet v1 [12]. The biological function of these raspberry allergens is still unknown, and their protein sequence is highly variable within the same species [13]. A high degree of structural homology has been demonstrated between Fra a 1 (strawberry), Mal d 1 (apple) and Bet v 1 from birch [14,15] and between PR-10 proteins from different Rosaceae fruits [16]. Rub i 1 and Rub i 3 are mainly responsible for the raspberry allergies occurring among berry fruit consumers. Profilins are the most widespread allergens throughout the plant kingdom. They are concentrated in the fruit cells’ cytosol. The molecular mass of profilins is 12-15 kDa, and their molecular structure is highly conserved, with 70%–85% homology among different species [17]. Inhibition experiments with serum pools from patients with fruit allergies have demonstrated cross-reactivity among the profilins Pru a 4, Fra a 1, and Fra a 3 [18].

In the present study, we aimed to identify and determine the content of bioactive compounds such as anthocyanins and the analogues of the most common panallergen Bet v 1 and the profilin content in different raspberry cultivars from organic and conventional cultivation. In the present literature, there is a complete lack of information about bioactive compound contents and the allergenic potential of raspberry cultivars. We investigated to see if there was a link between the flavonoid (anthocyanin) content in fruits and their allergenic status. The main hypothesis of the presented manuscript was to determine whether anthocyanin levels could be a determinant of the level of allergenic protein concentrations in raspberry fruit. If there is a relationship between anthocyanin concentration and potentially allergenic factors. Importantly, the present experiment was conducted over two years to be sure that the obtained results were not just a random effect.

## 2. Materials and Methods

### 2.1. Origin of the Fruits

The experiment was carried out in 2013–2014. Three raspberry cultivars were used for the experiment: ‘Laszka’, ‘Glen Ample’ and ‘Polka’. The experiment was conducted on the products of private farms: two organic and two conventional. All data about the farms’ locations and the methods used for farm management, the kind and dose of fertilizers used, and the methods used for plant protection at the time of cultivation are presented in Table 1. Detailed information on the weather forecast (minimum and maximum temperatures, number of hours of sunshine per day and rainfall) in the experimental area are presented in Figure 1.

### 2.2. Plant Material Preparation

For chemical analysis, the fruits of all cultivars from each of the experimental farms were harvested early in the morning and immediately transported (in cooling boxes) to the laboratory. A total of 250 g of fruits per sample were used in the analyses. All samples were freeze-dried using a Labconco (2.5) freeze-dryer (Warsaw, Poland, −50 °C, pressure 0.100 mBa). After the freeze-drying process, the plant material was ground in a laboratory mill A-11 (IKA^®^, Königswinter, Germany). The ground samples were then stored at −80 °C to avoid bioactive compound losses.

### 2.3. Polyphenols Separation and Identification

Polyphenols were measured by HPLC using a previously described method [19]. In brief: 0.100 mg of freeze-dried raspberry powder was extracted with 5 mL of 80% methanol with HPLC purity. Samples were mixed on Vortex 326 M (Marki, Poland). Next, samples were put into sonic bath Polsonic-3 (Warsaw, Poland) with parameters: 10 min, 30 °C, 5.5 kHz). After extraction samples were centrifuged (10 min, 3780× *g*, 5 °C). Obtained supernatant was collected to HPLC-vials and 100 µL of extract was inject into Phenomenx Fusion 80-A (C-18) 4.6 × 250 mm column (Warsaw, Poland). HPLC set: two pumps LC-20AD, controller CBM-20A, column oven SIL-20AC, spectrometer UV/Vis SPD-20 AV. Polyphenols were separated under gradient conditions with a flow rate of 1 mL/min. Two gradient phases were used: 10% (*v:v*) acetonitrile and ultrapure water (phase A) and 55% (*v:v*) acetonitrile and ultrapure water (phase B). The phases were acidified by orthophosphoric acid (pH 3.0). The total time of the analysis was 38 min. The phase-time program was as follows: 1.00–22.99 min, 95% phase A and 5% phase B; 23.00–27.99 min, 50% phase A and 50% phase B; 28.00–28.99 min, 80% phase A and 20% phase B; and 29.00–38.00 min, 95% phase A and 5% phase B. The wavelengths were 250–370 nm for all polyhenolic compounds. Bioactive compounds were identified by using 99.9% pure standards (Sigma-Aldrich, Warsaw, Poland) and the retention times for the standards.

### 2.4. Anthocyanin Separation and Identification

The first step of sample purification for the anthocyanin analysis was combined with the analysis of the polyphenols. The samples were extracted with 80% methanol. After the first centrifugation (see the previous section), 2.5 mL of supernatant was collected into a new plastic tube, and then 2.5 mL of 10 mol HCl and 5 mL of 100% methanol were added. The samples were gently shaken and put in a refrigerator (5 °C, 10 min). Next, 1 mL of extract was transferred into HPLC vials and analyzed. The anthocyanins were separated under isocratic conditions with a flow rate of 1.5 mL/min. One mobile phase, 5% acetic acid, methanol and acetonitrile (70:10:20), was used. HPLC set was performed from modules: two pumps (LC-20AD), one controller (CBM-20A), column oven (SIL-20AC), one spectrometer (UV/Vis SPD-20 AV). Phenomenx Fusion 80-A (4.6 × 250 mm, practical shape 4 µm) column (C18) was used. The analysis time was 10 min at a wavelength of 570 nm. The anthocyanins were identified by using 99.9% pure standards (Sigma-Aldrich) and the retention times for the standards [20].

### 2.5. Allergenic Potential Analysis

To obtain the proteins from the fruit, a Total Protein Extraction Kit for Plant Tissues (Sigma-Aldrich, Poland) was used. The analyses were performed according to the protocols described in a previous paper [19]. Their potential allergenicity was determined by indirect, non-competitive ELISA. For the primary antibodies, mouse antibodies against Bet v1 (Dendritics, Lion, France) for the detection of Bet v 1 analogues were used. Rabbit antibodies against profilin (Dendritics) for the determination of proteins such as profilin were also used. For the secondary antibodies, a conjugate of antibodies against mouse immunoglobulins with alkaline phosphatase (Sigma-Aldrich) or antibodies against the rabbit immunoglobulin conjugate with alkaline phosphatase (Sigma-Aldrich) were used. The wells on the plates were blocked with a 3% solution of commercial skim milk. As the substrate for alkaline phosphatase, pNPP (*p*-Nitrophenyl Phosphate (pNPP, KR-pT-IRR, Ser/Thr Phosphatase Assay Kit, BioAssay™, Sigma-Aldrich) was used, 3 M NaOH (Sigma-Aldrich) was used as the stopping reagent, and as a washing agent, PBS with 0.1% Tween 20 (Sigma-Aldrich) were used. The absorbance was read at 405 nm with the use of a Multiscan RC microplate reader (Labsystems, Helsinki, Finland), and the results were calculated using a standard curve prepared with the Bet v 1 allergen or profilin.

### 2.6. Statistical Analysis

The results obtained from the chemical analyses were statistically analyzed using Statgraphics Centurion 15.2.11.0 software (StatPoint Technologies, Inc., Warranton, VA, USA). The values presented in the tables are expressed as the mean values for the organic and conventional cultivation systems for the three raspberry cultivars ‘Laszka’, ‘Glen Ample’ and ‘Polka’. The statistical calculations were based on a two-way analysis of variance using Tukey’s test (*p* = 0.05). A lack of statistically significant differences between the examined groups is indicated by labelling with the same letters. A standard error (SE) is given for each mean value reported in the tables. To obtain a better picture of the correlation between the identified biologically active compounds and the allergenic proteins, a Principle Component Analysis (PCA) was used. The PCA figures were made using XLStat Trial version (Microsoft Excel, Chicago, IL, USA). Another correlation was demonstrated by calculating Pearson’s coefficients.

## 3. Results

The obtained results showed that the conventionally raised raspberries were characterized by a significantly (*p* < 0.0001) higher concentration of Bet v1 homologues than the organic ones, but only in 2013. In the next year of the experiment, we did not detect any difference between the raspberry fruits from the two cultivation systems regarding Bet v1. In 2014, the organic raspberries were characterized by 819.9 ng/g DW, and the conventional raspberries contained by 822.9 ng/g DW of Bet v1 homologues (Table 2 and Table 3). It is interesting that in both years, ‘Polka’ cv. was characterized by the lowest concentration of Bet v1 among the examined raspberry cultivars. The reverse situation was observed for the profilins. In 2013, there were no differences between the organic and conventional raspberries, whereas in 2014, the conventional fruits contained significantly more (*p* < 0.0001) profilins.

In both years of the experiment, ‘Laszka’ cv. was characterized by a significant (*p* < 0.0001) level of profilins at 4.47 μg/g DW and 5.98 μg/g DW in 2013 and 2014, respectively. As shown in Table 2 and Table 3, in both years, the conventional raspberries contained significantly more total polyphenols (*p* = 0.0004 and *p* < 0.0001) than the organic ones. In 2013, the highest level of total polyphenols was found in ‘Glen Ample’ cv.; however, in 2014, it was the ‘Polka’ cv. fruits. The content of phenolic acids was dependent on the cultivation system and cultivar only in 2014. The conventional raspberries contained significantly more total phenolic acids, with a value of 416.0 mg/100 g DW. The highest concentration of total phenolic acid was observed in 2014 for ‘Polka’ cv. (496.8 mg/100 g DW). In 2013, we did not observe any effect of the farming system on the gallic acid content. Only in 2014 were the conventional raspberries characterized by significant (*p* = 0.0031) concentration of gallic acid

The examined cultivars had differences in their gallic acid content in the raspberry fruits. In 2013, the highest level of gallic acid was noticed in ‘Polka’ cv., but in 2014, the highest level was in ‘Glen Ample’ cv. (Table 2 and Table 3). The content of chlorogenic acid was significantly higher in the conventional raspberry, though only in 2013. In both years of the experiment, ‘Glen Ample’ cv. was characterized by the highest level of chlorogenic acid in the fruits, with levels of 27.9 mg/100 g DW and 52.1 mg/100 g DW, respectively, in 2013 and 2014. The content of caffeic acid was quite variable between the experimental years. In 2013, the highest concentration of that compound was found in the organic raspberries, but in 2014, the concentration was higher in the conventionally raised raspberries. A similar situation was found for the effect of the cultivars. In 2013, in the ‘Polka’ cv. fruits, we found a significantly (*p* < 0.0001) higher concentration of caffeic acid, but in 2014, the concentration was highest in the ‘Laszka’ cv. fruits (*p* < 0.000). In 2013, there were no differences in *p*-coumaric acid content between the organic and conventional raspberry fruits. In 2014, the organic raspberries contained significantly more (8.14 mg/100 g DW) *p*-coumaric acid than the conventional ones (5.65 mg/100 g DW). The highest level of ***p***-coumaric acid was found for ‘Polka’ cv. in 2013 and for ‘Laszka’ cv. in 2014 (Table 2 and Table 3). The content of ferulic acid was only significantly higher in the conventional raspberry in 2013. We did not see any effect of the farming system or cultivar on the ferulic acid content in 2014. In the case of ellagic acid, there were no differences in 2013; in 2014, the conventional raspberries contained significantly more (*p* < 0.0001) phenolic acid, and the ‘Polka’ cv. fruits contained significantly more (*p* < 0.0001) ellagic acid. In the case of total flavonoids, we observed that in both years of the experiment, the conventional raspberry contained significantly more (*p* < 0.0001) of these compounds than the organic ones. Only in 2013 did the cultivar ‘Glen Ample’ produce a significantly higher concentration of total flavonoids in the fruits (819.5 mg/100 g DW). The organic raspberries contained significantly more total flavonols (*p* = 0.0027) but only in 2013. In the second year of the experiment, conventional raspberries were characterized by a higher concentration of total flavonols (*p* < 0.0001). In both years of the experiment, we observed that the conventional raspberry contained significantly more quercetin-3-O-rutinoside, with values of 8.83 mg/100 g DW and 14.10 mg/100 g DW, respectively, in 2013 and 2014. The cultivars had a significant effect on the quercetin derivative content. In 2013, the highest level of that compound was observed in ‘Glen Ample’ cv. fruits, but in 2014, the highest concentration was in ‘Polka’ cv. fruits (Table 2 and Table 3). In both years of the experiment, the content of myricetin was significantly higher in the organic raspberries. Only in 2013 did we observe any effect of the cultivar on the myricetin content. Organic raspberries were characterized by a higher level of luteolin, but only in 2013. In the next year, we observed that conventional fruits contained significantly (*p* < 0.0001) more luteolin (Table 2 and Table 3). The content of quercetin-3-O-glucoside and kaempferol were significantly higher in the conventional raspberries only in 2014. The total anthocyanins and two of three individual compounds were significantly higher in conventional fruits in both years of the experiment. The effect of the cultivar was noticed only in 2013. ‘Glen Ample’ cv. fruits contained significantly more cyanidin and pelargonidin than the rest of the examined raspberry cultivars. The current experiment provides information about the impact of cultivation method (organic and conventional) on the quality of raspberries. The authors have applied a holistic “from farm to fork” approach. The presented experiment not only shows the nutritional value and the content of the bioactive compounds in different raspberry cultivars but also demonstrates how raspberries can affect consumers who suffer from allergy problems. The individual ELISA results are presented in Table 4 and Table 5

The PCA results showed that the overall degree of variability explained by PC1 and PC2 was 94.54% in 2013 and 80.24% in 2013 and 2014, respectively (Figure 2 and Figure 3). This was confirmed by a strong link between the measured chemical compounds and the allergenic proteins identified in the raspberries. In both years of the experiment, the ‘Polka’ and ‘Laszka’ cultivars were positively correlated with organic raspberries. As shown in the graph, for both years of the experiment, the organic and conventional raspberries were grown in completely separate areas. This arrangement suggests a complete chemical dissimilarity between the examined fruits.

## 4. Discussion

Raspberries are among the most tasty and healthiest fruits in the world. Consumers like them and consume them in different forms, as both fresh and processed fruit. From a health point of view, raspberry fruits contain many bioactive compounds, such as polyphenols (especially flavonols and anthocyanins) and vitamin C [20,21,22,23]. According to strict farming rules, organic farming is one of the best alternatives for high-quality and safe fruit production [24]. Many experiments have shown that organic fruits contain more bioactive compounds than conventionally raised fruits [25,26,27], but some experiments have shown the opposite results [28,29]. In the present study, we showed that organic raspberries contained fewer total polyphenols and flavonoids, especially purple pigments (anthocyanins), than conventionally raised raspberries (Table 2 and Table 3). This finding is due to the use of dry matter content in fruits. Many results of chemical composition and polyphenolic compounds in raspberry fruits are presented as the fresh weight [30,31]. In such a situation, the higher dry matter and polyphenol content in organic fruits can lead to the opposite result—a lower concentration of phenolic compounds after the re-calculation of the results and presenting them as the dry matter content [26,32]. When a raspberry is consumed fresh or in desserts, the content of bioactive compounds in the fresh matter is the most important information for a consumer. On the other hand, we were looking for a link between the content of biologically active compounds and their allergenic potential. In this situation, we decided it was more informative to present the results calculated on a dry matter basis. Conventional raspberries contained more phenolic acids than organic raspberries. Similar results were presented by others [26,27]. The higher content of phenolic acids in conventional raspberry fruits may be an effect of abiotic salinity stress. It worth noting that in organic farming, only animal manure is used for fertilization. In conventional systems, mineral fertilizers are used. According to the data presented in Table 1, soil from the conventional farms had a higher salt concentration, which was reflected by the higher EC status. This could be the result of the effect of salinity stress inducing a higher phenolic acid production by the plants [33]. When plants are exposed to a higher salt concentration, their tissues produce more phenolic acids. These compounds are produced in the roots and then transported to the rest of the plant, including the fruits [34]. In both years of the experiment, we observed the lowest level of total phenolic acids in the ‘Laszka’ cv cultivar. The variation in phenolic acid contents between cultivars is based on genetic factors. In the experiment presented by Pavlović et al. [35] among four raspberry cultivars, ‘Tulameen’ cv. was characterized by the lowest level of phenolic acids, with a value of 604.6 mg/100 g DW, and ‘Willamette’ had the highest level (1021.4 mg/100 g DW). In raspberry fruits, flavonoids play an important role. Anthocyanins belong to the flavonoids group. They are synthesized and stored in raspberry fruits. Among the different plant secondary metabolites, flavonoids are important compounds. Plants produce flavonoids as a reaction against intensive sun radiation to protect their tissues against UV radiation [36]. We observed a stable reaction of raspberry fruits against sun radiation among the examined cultivars. In both years of the experiment, ‘Polka’ cv. contained significantly higher levels of total flavonols in its fruits (Table 2 and Table 3). A similar situation was observed in another experiment. Two different raspberry cultivars produced significantly different levels of total flavonols, ‘Amira’ cv. at 19.52 mg/100 g DW while ‘Polka’ cv. was 31.06 mg/100 g DW [22]. This effect could be explained by the genetic differentiation of raspberry cultivars. Anthocyanin synthesis in raspberry fruits depends on many intrinsic and external factors. There are genes that regulate anthocyanin synthesis originally at the transcriptional level. In addition, since these pigments provide plants with UV tolerance, their production is regulated by distinct special genes that are strongly regulated by sunlight exposure. A higher dose of light available for raspberry plants results in a higher anthocyanin concentration in the fruits. It should be noted that the longer exposure of raspberry plants to sunlight and a higher total number of sunny hours per day result in a higher concentration of the total anthocyanins in the raspberry fruits [37]. In the present experiment, we noted that phenomenon in action. According to the data presented in Figure 1, although the experimental farms were located very close to each other, the conventional area was in a higher sun zone than the organic area. In May and June, the conventional raspberry farm was exposed to higher levels of sunlight during the fruit set and development period than the organic farm. That is why in both years of the experiment, we found that the conventional fruits had higher amounts of total anthocyanins as well as individual purple pigments (cyanidin-3-*O*-glucoside and pelargonidin-3-*O*-glucoside) than the organic fruits (Table 2 and Table 3). We found a strong linear correlation between the total anthocyanins and Bet v1 content in 2014 as well as the profilins content in both experimental years for conventional raspberries (Table 6). Strawberry allergen has an impact on the pathway for the synthesis of enzymes responsible for the synthesis of anthocyanins. Therefore, a lower allergen content directly affects a lower anthocyanin content. White strawberries—colorless strawberry mutants known to be tolerated by individuals affected by allergies—were found to be virtually free from the strawberry allergen [38]. It seems that the white fruits were more easily tolerated by volunteers suffering from fruit allergies than normal fruits [39]. Moreover, the level of Bet v1 in white strawberry mutants was almost zero. This could mean that dark-colored strawberries increase patients’ allergic reactions. Because raspberries and strawberries are closely related (the same family Rosaceae), we could expect similar results with raspberries. In our experiment, the level of Bet v1 was higher in the conventional raspberries (Table 2). Among the patients suffering from specific Rosaceae fruit allergies, it is not possible to evaluate which fruit and pollen allergies are typically associated from the available studies. In one of these studies, carried out in Spain, profilin-sensitized patients allergic to Rosaceae fruits and pollens were sensitized to both a higher number of fruits and a wider variety of fruits, including fruits outside the Rosaceae family [12]. In Europe, Rosaceae fruit allergies and LTP sensitization are almost absent. The major fruit allergy pattern characteristic of this area is an association between birch pollen allergy and Rosaceae fruit allergy, with PR-10 as the most frequent cross- reacting allergen [40]. In addition, we should remember that conventionally raised raspberry fruits may contain many harmful chemicals and compounds. Among the synthetic chemicals used in conventional agriculture, pesticides are the most serious. These chemicals can have carcinogenic, neurodegenerative and allergic effects [41]. Crops produced under the strict rules of organic farming are safer and contain fewer toxins as well as fewer allergenic substances than products from conventional farming. The present results are in accordance with the beliefs of consumers, who feel that organic fruits are safer and of higher quality. It is worth pointing out that in the case of raspberries, we found lower levels of dangerous Bet v1 and profilins in the organic fruits (Table 2 and Table 3). However, the obtained results are opposite of findings presented previously for different organic crops. Organic apricots were characterized by higher Bet v1 and profilin contents [19]. Organic tomato contained higher concentrations of profilins but not Bet v1 [42]. There is also some evidence that fully ripened tomato fruits cause a stronger allergic reaction than orange red fruits (not fully ripened) [43]. In tomatoes, it seems that the allergenic potential is connected with the pigment carotenoids. Of course, in the examined raspberries, we found a link between pigments and allergy potential, but anthocyanins and carotenoids belong to completely different chemical groups. On the other hand, it could be pointed out that allergic reactions may involve exposure to different chemical agents. In the case of organic tomatoes, it was not only food intolerance but also a skin test that confirmed the higher level of their allergenic potential. Using genetic and breeding approaches, it is possible to find numerous Rubus genotypes for the Rub i 1 and Rub i 3 proteins. In could be that some wild varieties or plants growing in less sunny areas (the north Europe countries) seem to be free of allergens. Thus, farming practices can take advantage of the biodiversity of Rubus to select for hypoallergenic raspberry lines [44]. On the other hand, based on the obtained results, we may take advantage of the fact that organic raspberries contain a lower level of Bet v1.

## 5. Conclusions

Considering the obtained results, organic raspberry could be more useful for those consumers suffering from slight allergy symptoms. The higher level of anthocyanins could be used as an indicator for the concentration of allergy proteins.

## Figures and Tables

**Figure 1 antioxidants-09-00256-f001:**
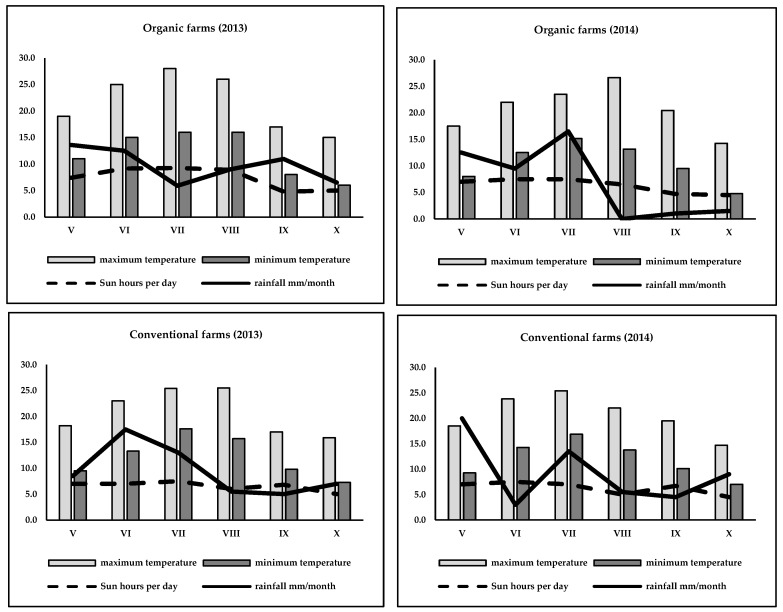
Weather conditions in experimental farms (organic and conventional) 2013–2014 in time of raspberry fruits development.

**Figure 2 antioxidants-09-00256-f002:**
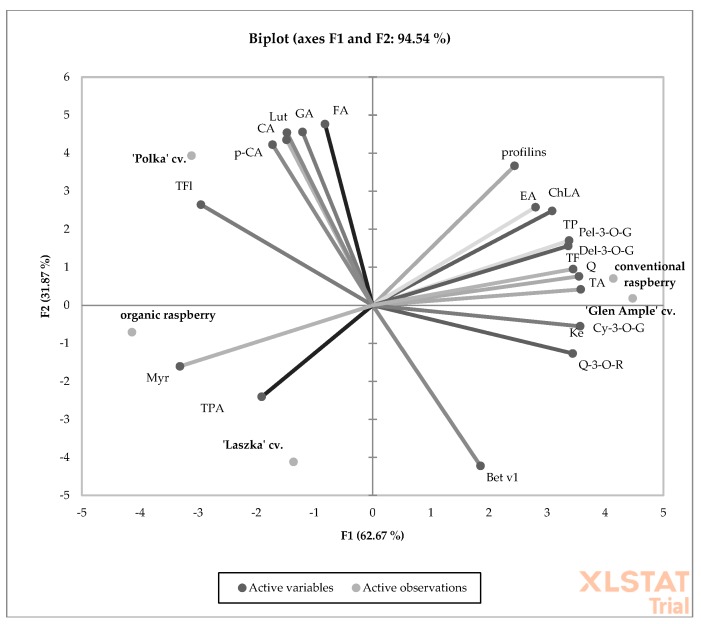
PCA analysis showing the relationship between the chemical composition and allergy potential of organic and conventional raspberry in 2013. (Bet v1) Bet v1; (profilins) profilins; (TP) total polyphenols; (TPA) total phenolic acids; (GA) gallic acid; (ChLA) chlorogenic acid; (CA) caffeic acid; (*p*-CA) *p*-coumaric acid; (FA) ferulic acid; (EA) ellagic acid; (TF) total flavonoids; (TFl) total flavonols; (Q-3-O-R) quercetin-3-O-rutinoside; (Myr) myricetin; (Lut) luteolin; (Q) quercetin; (Ke) kaempferol; (TA) total anthocyanins; (Cy-3-O-G) cyanidin-3-O-glucoside; (Pel-3-O-G) pelargonidoin-3-O-glucoside; (Del-3-O-G)delphinidin-3-O-glucoside.

**Figure 3 antioxidants-09-00256-f003:**
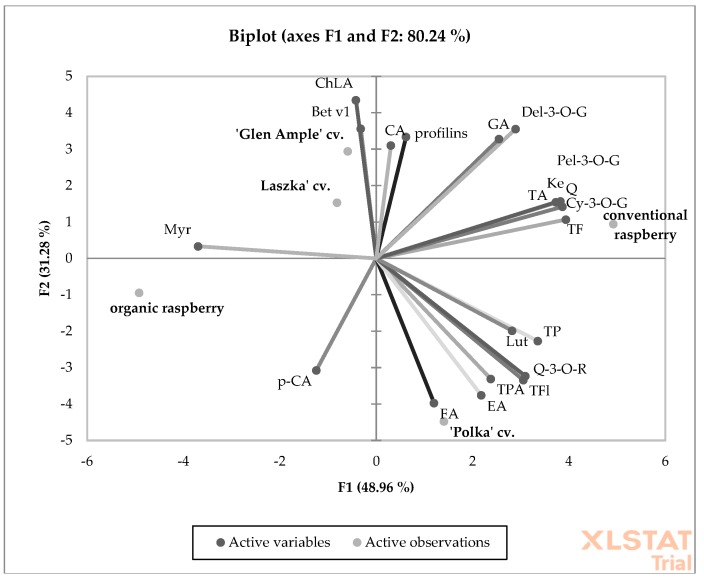
PCA analysis showing the relationship between the chemical composition and allergy potential of organic and conventional raspberry in 2014. (Bet v1) Bet v1; (profilins) profilins; (TP) total polyphenols; (TPA) total phenolic acids; (GA) gallic acid; (ChLA) chlorogenic acid; (CA) caffeic acid; (*p*-CA) *p*-coumaric acid; (FA) ferulic acid; (EA) ellagic acid; (TF) total flavonoids; (TFl) total flavonols; (Q-3-O-R) quercetin-3-O-rutinoside; (Myr) myricetin; (Lut) luteolin; (Q) quercetin; (Ke) kaempferol; (TA) total anthocyanins; (Cy-3-O-G) cyanidin-3-O-glucoside; (Pel-3-O-G) pelargonidoin-3-O-glucoside; (Del-3-O-G)delphinidin-3-O-glucoside.

**Table 1 antioxidants-09-00256-t001:** Characterization of localization, fertilizers regime and plant protection used for organic and conventional raspberry cultivation in (2013-2014).

Cultivation System	Localization	Type of Soil	Kind of Fertilizer	Dose of Fertilizers and Time of Given	Plant Protection System
Organic farm no. 1 (*n* = 6)	Zakroczym	sandy middle soil IVa and IVb category (15% floatable particles) pH (5.5), EC (3.8)	cow manure	35 t ha-1 one year before raspberry planting	Grevit 200 SL
	(52°26″ N 20°36″ E)				
Organic farm no. 2 (*n* = 6)	Załuski	sandy middle soil, sandy-clay IV category (20% floatable particles), pH (5.5), EC (4.1)	cow manure	30 t ha-1 one year before raspberry planting	no protection
	(52°37″ N 20°22″ E)				
Conventional farm no. 1 (*n* = 6)	Czerwińsk nad Wisłą	sandy-loamy middle soil IV and III category (20% floatable particles), pH (5.5), EC (5.2)	Hydrocomplex 12-11-18; Superba 8-11-36	(200 kg ha-1, 150 kg ha-1) in autumn a year before raspberry planting; 3 doses in time of cultivation	Signum 33 WG, Miros 20 SP,
	(52°23″ N 20°20″ E)				
Conventional farm no. 2 (*n* = 6)	Czerwińsk nad Wisłą	sandy-loamy middle soil IV and III category (25% floatable particles), pH (5.5), EC (5.5)	amonium nitrate, polyphosphate, magnesium sulphate	in autumn a year before raspberry planting; 3 doses in time of cultivation	Calypso 480 SC, Miros 20 SP, Zato 50 WG
	(52°23″ N 20°20″ E)				

**Table 2 antioxidants-09-00256-t002:** The content of allergenic analogs and polyphenols in examined raspberry cultivars from organic and conventional farming system in 2013.

Bioactive Compounds	Organic Raspberry	Conventional Raspberry	‘Laszka’ cv.	‘Glen Ample’ cv	‘Polka’ cv.	*p*-Value	
						System	Cultivar
Bet v1 (µg/g DW)	786.40 ± 27.00 ^1^ B ^2^	864.69 ± 52.15 A	917.88 ± 19.58 a	850.52 ± 47.16 ab	708.24 ± 27.07 b	<0.0001	<0.0001
Profilins (µg/g DW)	3.48 ± 0.38 A	3.49 ± 0.34 A	4.47 ± 0.56 a	2.44 ± 0.27 c	3.55 ± 0.80 b	N.S.	<0.0001
Total polyphenols (mg/100 g DW)	1009.84 ± 0.55 B	1172.36 ± 0.35A	1029.83 ± 0.10 b	1172.41 ± 0.08 a	1071.07 ± 0.23 a	0.0004	0.0135
Total phenolic acids	313.23 ± 2.16 A	359.33 ± 4.41 A	307.39 ± 1.47 a	352.85 ± 3.87 a	348.61 ± 0.47 a	N.S.	N.S.
Gallic acid	2.24 ± 0.61 A	2.46 ± 0.63 A	1.49 ± 0.09 b	1.80 ± 0.39 b	3.75 ± 0.10 a	N.S.	<0.0001
Chlorogenic acid	14.97 ± 3.39 B	26.15 ± 2.73 A	13.56 ± 0.43 b	27.91 ± 0.50 a	20.22 ± 0.54 ab	<0.0001	<0.0001
Caffeic acid	3.55 ± 0.78 A	2.27 ± 0.49 B	1.19 ± 0.12 c	2.84 ± 0.31 b	4.70 ± 0.15 a	<0.0001	<0.0001
*p*-Coumaric acid	13.92 ± 8.08 A	13.43 ± 8.24 A	8.29 ± 3.31 b	8.65 ± 3.86 b	24.08 ± 6.59 a	N.S.	<0.0001
Ferulic acid	5.23 ± 1.35 B	5.78 ± 1.67 A	4.10 ± 1.39 b	4.91 ± 1.08 b	7.49 ± 1.36 a	0.0350	<0.0001
Ellagic acid	273.49 ± 0.13 A	307.86 ± 0.57 A	280.74 ± 0.22 a	296.94 ± 0.40 a	294.33 ± 0.15 a	N.S.	N.S.
Total flavonoids (mg/100 g DW)	696.61 ± 2.54 B	813.04 ± 1.35 A	722.45 ± 3.04 b	819.56 ± 1.33 a	722.46 ± 1.19 b	<0.0001	0.0002
Total flavonols (mg/100 g DW)	55.42 ± 2.68 A	43.24 ± 2.91 B	45.13 ± 2.34 b	45.28 ± 1.79 b	57.57 ± 1.96 a	0.0027	0.0050
Quercetin-3-*O*-rutinoside	7.85 ± 1.93 B	8.83 ± 2.21 A	8.30 ± 0.56 b	9.09 ± 0.55 a	7.63 ± 0.43 c	0.0016	0.0013
Myricetin	26.60 ± 0.68 A	15.23 ± 0.55 B	23.73 ± 0.08 a	17.31 ±0.31 b	21.72 ± 0.23 ab	<0.0001	0.0080
Luteolin	11.87 ± 1.00 A	8.93 ± 1.17 B	4.86 ± 1.14 c	9.39 ± 0.66 b	16.96 ± 0.23 a	0.0001	<0.0001
Quercetin-3-*O*-glucoside	4.37 ± 0.87 A	4.73 ± 0.32 A	2.87 ± 0.68 c	4.75 ± 0.56 b	6.03 ± 0.40 a	N.S.	<0.0001
Kaempferol	4.72 ± 17.34 A	5.51 ± 1.68 A	5.37 ± 1.46 a	4.74 ± 1.19 a	5.24 ± 1.78 a	N.S.	N.S.
Total anthocyanins (mg/100 g DW)	641.20 ± 0.85 B	769.80 ± 10.42 A	677.31 ± 2.93 b	774.28 ± 11.74 a	664.89 ± 5.47 b	<0.0001	0.0001
Cyanidin-3-*O*-glucoside	320.21 ± 17.34 B	380.58 ± 15.68 A	344.49 ± 3.46 ab	384.90 ± 13.19 a	321.79 ± 16.78 b	0.0006	0.0060
Pelargonidin-3-*O*-glucoside	95.47 ± 0.85 B	141.44 ± 10.42 A	106.05 ± 2.93 b	135.16 ± 11.74 a	114.16 ± 5.47 ab	<0.0001	<0.0001
Delphinidinn-3-*O*-glucoside	225.51 ± 14.36 A	247.78 ± 15.36 A	226.77 ± 6.20 b	254.22 ± 8.38 a	228.94 ± 14.66 b	N.S.	<0.0001

^1^ Data are presented as the mean ± SE with ANOVA *p*-value; ^2^ Means in rows followed by the same letter (A, B, a–c) are not significantly different at the 5% level of probability (*p* < 0.05); N.S. not significant; (n) number of samples (field replications) *n* = 36 for system, *n* = 12 of cultivar.

**Table 3 antioxidants-09-00256-t003:** The content of allergenic analogs and polyphenols in examined raspberry cultivars from organic and conventional farming system in 2014.

Bioactive Compounds	Organic Raspberry	Conventional Raspberry	‘Laszka’ cv.	‘Glen Ample’ cv.	‘Polka’ cv.	*p*-Value	
						System	Cultivar
Bet v1 (µg/g DW)	819.98 ± 34.59 ^1^ A ^2^	822.94 ± 95.30 A	804.42 ± 22.45 b	861.73 ± 7.64 a	798.22 ± 88.61 b	N.S.	0.0002
Profilins (µg/g DW)	3.40 ± 0.16 B	4.71 ± 0.58 A	5.98 ± 0.70 a	3.68 ± 0.41 b	2.50 ± 0.53 c	<0.0001	<0.0001
Total polyphenols (mg/100 g DW)	835.33 ± 0.07 B	1067.16 ± 0.05 A	836.18 ± 0.04 c	931.41 ± 0.02 b	1086.15 ± 0.07 a	<0.0001	<0.0001
Total phenolic acids	309.53 ± 9.67 B	416.00 ± 4.48 A	244.20 ± 2.14 c	347.27 ± 4.01 b	496.82 ± 6.69 a	<0.0001	<0.0001
Gallic acid	0.73 ± 0.05 B	0.94 ± 0.23 A	0.79 ± 0.13 b	0.94 ± 0.03 a	0.77 ± 0.02 b	0.0031	0.0480
Chlorogenic acid	39.56 ± 1.43 A	41.02 ± 0.56 A	38.03 ± 0.56 b	52.16 ± 0.09 a	30.67 ± 0.65 b	N.S.	0.0002
Caffeic acid	1.13 ± 0.14 B	1.33 ± 0.44 A	1.65 ± 0.09 a	1.13 ± 0.04 ab	0.92 ± 0.36 b	0.0001	<0.0001
*p*-Coumaric acid	8.14 ± 9.97 A	5.65 ± 80.20 B	8.84 ± 9.88 a	3.54 ± 2.45 b	8.31 ± 49.89 a	<0.0001	<0.0001
Ferulic acid	1.49 ± 0.19 A	1.59 ± 0.70 A	1.63 ± 0.71 a	1.13 ± 0.19 a	1.85 ± 0.03 a	N.S.	N.S.
Ellagic acid	276.94 ± 0.81 B	365.66 ± 3.99 A	211.36 ± 0.52 b	288.37 ± 0.73 b	464.18 ± 3.32 a	<0.0001	<0.0001
Total flavonoids (mg/100 g DW)	525.80 ± 0.09 B	651.16 ± 0.12 A	591.98 ± 0.10 a	584.14 ± 0.03 a	589.32 ± 0.09 a	<0.0001	N.S.
Total flavonols (mg/100 g DW)	23.84 ± 1.32 B	32.50 ± 5.32 A	25.45 ± 0.27 ab	23.88 ± 0.77 b	35.17 ± 4.51 a	<0.0001	<0.0001
Quercetin-3-*O*-rutinoside	6.92 ± 0.09 B	14.10 ± 0.23 A	9.10 ± 0.03 b	6.58 ± 0.16 c	15.85 ± 0.10 a	<0.0001	<0.0001
Myricetin	2.77 ± 0.14 A	2.51 ± 0.40 B	2.60 ± 0.11 a	2.71 ± 0.09 a	2.60 ± 0.30 a	0.0330	N.S.
Luteolin	1.07 ± 0.50 B	1.59 ± 0.95 A	0.90 ± 0.33 b	1.41 ± 0.16 ab	1.68 ± 0.83 a	<0.0001	<0.0001
Quercetin-3-*O*-glucoside	1.85 ± 0.11 B	2.09 ± 0.90 A	1.72 ± 0.57 b	1.76 ± 0.39 b	2.44 ± 0.56 a	0.0078	N.S.
Kaempferol	11.22 ± 2.51 B	12.21 ± 2.05 A	11.14 ± 2.32 b	11.41 ± 1.22 b	12.60 ± 1.13 a	0.0025	0.0014
Total anthocyanins (mg/100 g DW)	501.96 ± 2.25 B	618.66 ± 2.67 A	566.53 ± 3.35 a	560.26 ± 2.46 a	554.15 ± 3.06 a	<0.0001	N.S.
Cyanidin-3-*O*-glucoside	260.54 ± 7.51 B	354.68 ± 4.05 A	311.39 ± 12.32 a	307.01 ± 11.22 a	304.43 ± 18.13 a	<0.0001	N.S.
Pelargonidin-3-*O*-glucoside	55.24 ± 2.25 B	72.24 ± 2.67 A	65.82 ± 3.35 a	62.79 ± 2.46 a	62.61 ± 3.06 a	0.0003	N.S.
Delphinidinn-3-*O*-glucoside	186.18 ± 6.23 A	191.74 ± 4.17 A	189.32 ± 3.96 a	190.46 ±1.11 a	187.10 ± 5.16 a	N.S.	N.S.

^1^ Data are presented as the mean ± SE with ANOVA *p*-value; ^2^ Means in rows followed by the same letter (A, B, a–c) are not significantly different at the 5% level of probability (*p* < 0.05); N.S. not significant; (n) number of samples (field replications). *n* = 36 for system, *n* = 12 of cultivar.

**Table 4 antioxidants-09-00256-t004:** The content of Bet v1 as ELISA results ng/g DW.

Organic Raspberry 2013			Conventional Raspberry 2013		
‘Laszka’ cv.	‘Glen Ample’ cv.	‘Polka’ cv.	‘Laszka’ cv.	‘Glen Ample’ cv.	‘Polka’ cv.
920.20	769.20	667.10	918.50	934.50	742.70
890.30	779.40	687.11	899.70	935.60	752.30
944.10	759.90	660.30	934.50	924.50	739.90
**Organic Raspberry 2014**			**Conventional Raspberry 2014**		
‘Laszka’ cv.	‘Glen Ample’ cv.	‘Polka’ cv.	‘Laszka’ cv.	‘Glen Ample’ cv.	‘Polka’ cv.
783.90	895.90	756.00	830.10	813.70	827.70
799.20	945.60	788.90	823.40	825.60	839.90
777.80	866.70	765.80	812.12	822.90	811.00

**Table 5 antioxidants-09-00256-t005:** The content of profilins as ELISA results µg/g DW.

Organic Raspberry 2013			Conventional Raspberry 2013		
‘Laszka’ cv.	‘Glen Ample’ cv.	‘Polka’ cv.	‘Laszka’ cv.	‘Glen Ample’ cv.	‘Polka’ cv.
6.263	3.767	3.56	2.483	1.214	6.712
6.759	2.682	3.62	2.864	1.790	5.922
6.099	3.434	3.46	2.323	1.777	6.331
**Organic Raspberry 2014**			**Conventional Raspberry 2014**		
‘Laszka’ cv.	‘Glen Ample’ cv.	‘Polka’ cv.	‘Laszka’ cv.	‘Glen Ample’ cv.	‘Polka’ cv.
3.740	2.284	4.741	8.139	5.104	2.54
3.705	2.203	4.153	8.981	4.927	2.39
3.323	2.345	4.099	8.009	5.244	2.57

**Table 6 antioxidants-09-00256-t006:** The value of R^2^ coefficient for Pearson regression between phenolics compounds and allergen analogs for organic and conventional raspberries.

Type of Regression	2013		2014	
	Organic Raspberry	Conventional Raspberry	Organic Raspberry	Conventional Raspberry
polyphenols/Bet v1	+0.8349	+0.7017	+0.6213	+0.7705
*p*-value	0.006	0.0048	0.0116	0.0019
polyphenols/profilins	+0.7934	+0.933	+0.8958	+0.8235
*p*-value	0.0130	<0.0001	0.0001	0.0007
anthocyanins/Bet v1	+0.6995	+0.5631	+0.8445	+0.9065
*p*-value	0.0050	0.019	0.0005	<0.0001
anthocyanins/profilins	+0.6781	+0.9852	+0.7650	+0.8235
*p*-value	0.0064	<0.0001	<0.0001	0.0002

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
