# Peer review of "The Interaction between Antioxidants Content and Allergenic Potency of Different Raspberry Cultivars"

_antioxidants, 2020, doi:10.3390/antiox9030256_

Round 1

Reviewer 1 Report

This article provides an interesting source of information about allergenic potency of different raspberry cultivars. However there are a few aspects that need revision.

Introduction

Introduction part is appropriate and contains adequate information regarding to this study.

Materials and Methods

Page 2, line 76, full stop is missing.

Page 4, line 104, 1.5 mL/min instead 1.5 mL m-1

In part 2.4 Anthocyanin separation and identification I would welcome some information about HPLC systém, especially type and size of chromatography column

page 4, line 109; page 10, line 228 – why are citations presented in different way? (Hallmann et al. 2018 or Fernandez et al. 2012 instead numbering)

Results

The results are exactly described, in terms of their importance, statistical and interaction analyses are presented.

Some mistakes in Table 2 and 3

myricetin instead myrycetin

quercetin-3-O-rutinoside is in tables two times with different values

space in front of and behind symbol ± is missing

in Table 2, line chlorogenic acid, column organic raspberry, symbol ± is missing

Discussion

Discussion of the results is exhaustive and no comments are necessary.

Page 11, line 279 misspeling in cyanidin-3-O-glucoiside

Conclusions

Conclusions are brief and clear

References

References are sufficient and coherent to the study, but format/style of reference is nonunifrom. Pleas check if referencing style is unified and according to the Instruction for Authors.

Author Response

Reviewer no. 1

Thank you very much for the review and for your recommendation to publish our manuscript in the “Antioxidant” journal.

Comment 1: “…Page 2, line 76, full stop is missing.…”

Authors’ response: As suggested by the Reviewer, full stop is missing was added at the end of the sentence.

Comment 2: “…Page 4, line 104, 1.5 mL/min instead 1.5 mL m-1

Authors’ response: As suggested by the Reviewer, not correct form 1.5 mL min-1  has been changed into 1.5 mL/min.

Comment 3: “…In part 2.4 Anthocyanin separation and identification I would welcome some information about HPLC systém, especially type and size of chromatography column

Authors’ response: As suggested by the Reviewer all missing data about HPLC set and parameter was added into section.

Comment 4: “…page 4, line 109; page 10, line 228 – why are citations presented in different way? (Hallmann et al. 2018 or Fernandez et al. 2012 instead numbering)

Authors’ response: Different method of the citation (author names, not numbers) was the result of authors inadvertent. The incorrect citation method has been corrected in the manuscript text.

Comment 5: “…Some mistakes in Table 2 and 3: myricetin instead myrycetin

Authors’ response: As suggested by the Reviewer incorrect name myrycetin was change into myrycetin in Tables, Figures and manuscript text.

Comment 6: “…quercetin-3-O-rutinoside is in tables two times with different values..”

Authors’ response: Thank you for pointing mistake. The second compound described as quercetin-3-O-rutinoside should be described as quercetin-3-O-glucoside. Incorrect (double name) of bioactive compounds was corrected in tables and the manuscript text.

Comment 7: “…space in front of and behind symbol ± is missing..”

Authors’ response: As suggested by the Reviewer all missing spaces in front of and behind symbol ±  in Tables 1 and 2 were corrected.

Comment 8: “…in Table 2, line chlorogenic acid, column organic raspberry, symbol ± is missing.

Authors’ response: As suggested by the Reviewer the missing mark “±” was added into right place.

Comment 9: “…Page 11, line 279 misspeling in cyanidin-3-O-glucoiside..”

Authors’ response: As suggested by the Reviewer misspelling in name of cyanidin-3-O-glucoside was corrected.

Comment 10: “…References are sufficient and coherent to the study, but format/style of reference is nonunifrom. Pleas check if referencing style is unified and according to the Instruction for Authors.

Authors’ response: As suggested by the Reviewer reference list was carefully checked according to Instruction for Authors. All mistakes were corrected

Reviewer 2 Report

Authors should provide more details regarding plant cultivation,  and the conditions of sampling. What soil type was used? In which air temperature and relative humidity plants grown.  All this kind of information in crucial because all of them greatly influence the metabolites content of plants and therefore, the bioactivity of plant extracts. This several information's may be included here and as supplementary information.

Please find my comments below:

Line 33. Use "full stop" after "abiotic environmental stresses [7] "

Line 54 - 57: Use references

Table 1, Table, table 3: Start each parameters with capital letters.

Fig. 1 is not clear. Use picture with higher resolutions.

Line 106: Replace "analysis times" with retention times"

Line 108: write the proper addresses of "total protein extraction kit"

Line 117: What is pNPP? Define it

Line 135: No need to write full form of same thing again and again.

   "PCA"

2.6: Remove some unwanted description in the statistical analysis section. For example line 123 to 140.

Table 2 and 3

In the footnotes " 1 data are presented-----"

where is 1? insert it in the table as well

Table 2 and 3

Replace "gallic" by "gallic acid"

"Chlorogenic" by "chlorogenic acid".... "caffieic" ......and so on

Fig. 3 is not clear. Please supply more clear figures.

Discussions: Avoid writing data values again in the discussions sections.

Line 256 to 258.......268 to 270.. References are missing.

Author Response

Reviewer no. 2

Thank you very much for the review and for your recommendation to publish our manuscript in the “Antioxidant” journal.

Comment 1: “…Line 33. Use "full stop" after "abiotic environmental stresses [7] ".…”

Authors’ response: According to Reviewer suggestion “full stop” after pointed place was written.

Comment 2: “…Line 54 - 57: Use references.…”

Authors’ response: According to Reviewer suggestion proper reference was used.

Comment 3: “…Table 1, Table, table 3: Start each parameters with capital letters..…”

Authors’ response: According to Reviewer suggestion all parameters in presented Tables now are written by capital letter. 

Comment 4: “…Fig. 1 is not clear. Use picture with higher resolutions..…”

Authors’ response: According to Reviewer suggestion Figure 1 resolution was corrected.

Comment 5: “…Line 106: Replace "analysis times" with retention times".…”

Authors’ response: According to Reviewer suggestion "analysis times"  was changed into  retention times".

Comment 6: “…Line 108: write the proper addresses of "total protein extraction kit".…”

Authors’ response: According to Reviewer suggestion a proper address of “total protein extraction kit” was added into manuscript text.

Comment 7: “…Line 117: What is pNPP? Define it.…”

Authors’ response: According to Reviewer suggestion full name of pNPP was defined in proper manuscript section.

Comment 8: “…Line 135: No need to write full form of same thing again and again.   "PCA".…”

Authors’ response:  According to Reviewer suggestion PCA was defined in full name only first time in section “Statistical analysis”. Next only abbreviation was used in the manuscript text.

Comment 9: “…2.6: Remove some unwanted description in the statistical analysis section. For example line 123 to 140..…”

Authors’ response: According to Reviewer suggestion all information about statistical analysis, which was as well presented under tables was removed from section.

Comment 10: “…Table 2 and 3 In the footnotes " 1 data are presented-----" where is 1? insert it in the table as well Table 2 and 3.…”

Authors’ response: The missing labels (1 and 2 numbers) in tables was added into proper places.

Comment 11: “…Replace "gallic" by "gallic acid" "Chlorogenic" by "chlorogenic acid".... "caffieic" ......and so on.…”

Authors’ response: According to Reviewer suggestion short names of phenolic acids was replaced by their full names.

Comment 12: “…Fig. 3 is not clear. Please supply more clear figures..…”

Authors’ response: According to Reviewer suggestion both figures (Figure 2 and 3) were re-prepared using other software. In their current form they are much more visible and clear.

Comment 13: “…Avoid writing data values again in the discussions sections.…”

Authors’ response: According to Reviewer suggestion all values ​​presented earlier in the tables have been removed from the Discussion section, to avoid double presentation.

Comment 14: “…Line 256 to 258.......268 to 270.. References are missing..…”

Authors’ response: According to Reviewer suggestion all missing references have been carefully checked and corrected.

Reviewer 3 Report

The study of Hallmann et al. presents studies related to the correlation between phenolic compound content and allergenic potential of raspberry cultivars, collected from organic and conventional farms. The results of the authors demonstrated that organically farmed raspberries are safer for the consumer due to their decreased allergic potential. This reduced allergenic activity was associated with the lower flavonoid content of the organic cultivars. Furthermore, an association between anthocyanin content and allergenic potential was suggested by the authors.

Overall, the manuscript is clearly presented and addresses an interesting subject. I have the following comments for the authors:

The conclusions of the study point to a link between anthocyanin content and the allergic potential of conventional raspberry cultivars.  However, as the authors stated, a correlation between anthocyanin levels and Bet v1 content was only established in 2014. This correlation was not observed in the previous year. Furthermore, Bet v1 and profilin levels were only increased in conventional fruit in one experimental year. Why didn’t the authors prolong their study to determine a stronger correlation and how can the link between anthocyanin levels and allergenic potential  be made in light of these inconsistencies?

I feel that there is  too little emphasis placed in the discussion on the allergenic potential of  raspberries and also on the association of anthocyanins in this process. The authors have not addressed a few studies related to the identification of raspberry allergens. Apart from Rub i 1 and Rub i 3, cyclophilin and a class III chitinase are proteins potentially contributing to the allergenicity of raspberries. Furthermore, the authors claim that Hjerno et al. 2006 concluded from their research that ‘anthocyanins could be responsible for causing allergic reactions’. This is not entirely correct , as Hjerno et al. suggested that there could be a possible cross regulation between anthocyanin  and allergen biosynthesis pathways. Furthermore, Bet v 1 could be a carrier of flavonoid-like compounds. This should be addressed.

The results of the ELISA analysis should be presented and addressed in the Results section.

Line 257: flavonoids do not belong to the anthocyanin group, but anthocyanins are classified into the group of flavonoids.

Author Response

Reviewer no. 3

Thank you very much for the review and for your recommendation to publish our manuscript in the “Antioxidant” journal.

Comment 1: “…The conclusions of the study point to a link between anthocyanin content and the allergic potential of conventional raspberry cultivars.  However, as the authors stated, a correlation between anthocyanin levels and Bet v1 content was only established in 2014. This correlation was not observed in the previous year. Furthermore, Bet v1 and profilin levels were only increased in conventional fruit in one experimental year. Why didn’t the authors prolong their study to determine a stronger correlation and how can the link between anthocyanin levels and allergenic potential  be made in light of these inconsistencies?.…”

Authors’ response: Presented results of  experiment was planned on a multi-years experiment. The first year was carried out in 2013. A set of research material was collected. This year, the correlation between anthocyanin content and allergenic factors was weak, but it occurred (Table 4). In 2014, a much stronger relationship was observed. However, in 2015 there were major complications on farms producing fruit for experimental purposes and, unfortunately, it was not possible to collect a sufficient sample of the material for analysis. Therefore, the experiment must be to interrupted. The authors agree with Reviewer remark that this type of research should be continued. Therefore, our results should be treated more as preliminary studies. The results of this research give a certain picture of dependence and will be continued in the future to determine the previously observed dependence.

Comment 2: “… I feel that there is  too little emphasis placed in the discussion on the allergenic potential of  raspberries and also on the association of anthocyanins in this process. The authors have not addressed a few studies related to the identification of raspberry allergens. Apart from Rub i 1 and Rub i 3, cyclophilin and a class III chitinase are proteins potentially contributing to the allergenicity of raspberries..…”

Authors’ response: Authors are grateful for the Reviewer for pointing out new research directions. Of course, a very interesting problem will be further identification of potentially allergenic factors (proteins as: cyclophilin and a class III chitinase as well specific raspberry allergens as Rub i1 and Rub i3) in raspberry fruits. However, the purpose of this work was not to identify all allergenic factors, but to provide a preliminary understanding of the potential allergenicity of raspberry fruit. The work focuses on the relationship between the Bet v1 homologue and anthocyanin compounds.

Comment 3: “…Furthermore, the authors claim that Hjerno et al. 2006 concluded from their research that ‘anthocyanins could be responsible for causing allergic reactions’. This is not entirely correct, as Hjerno et al. suggested that there could be a possible cross regulation between anthocyanin  and allergen biosynthesis pathways. Furthermore, Bet v 1 could be a carrier of flavonoid-like compounds. This should be addressed...…”

Authors’ response: Of course, Authors thank you very much for pointing out the correctness of the inference. Experiment of Hjernø et al. (2006) was the initial inspiration to apply presented research. As the authors emphasize, these are quite preliminary studies and in the future require further research and experimental directions. Strawberry fruit with reduced Fra a1 content are white fruit mutants. The relationship between the reduced content of Fra a1 and enzymes conditioning the synthesis of anthocyanin compounds in strawberry fruit was studied by Franz-Oberdorf et al. (2017). Fruits of white strawberries were much better tolerated by people with a slight allergy and food intolerance to these fruits.

Comment 4: “….The results of the ELISA analysis should be presented and addressed in the Results section..…”

Authors’ response: According to Reviewer suggestion results of ELISA analysis and their description was added to Results section.

Comment 5: “…Line 257: flavonoids do not belong to the anthocyanin group, but anthocyanins are classified into the group of flavonoids...…”

Authors’ response: According to Reviewer suggestion incorrectly formulated sentence has been corrected in the text of the maqnuscript.

Reviewer 4 Report

In the manuscript “The interaction between antioxidants content and allergenic potency of different raspberry cultivars” Hallmann et al. would like to link polyphenols and anthocyanin content with the allergy status of the conventional and organic raspberry fruits. The authors don’t give a hypothesis about the link between the antioxidant compounds content and the values of Bet v1 and profilin; in addition the data presented don’t show any correlation. In my opinion this work can not be published in the present form. In the Conclusions the authors said : “Considering these results along with the other beneficial aspects of organic cultivation, less environmental pollution and production of more controlled food, that this second solution is much more beneficial for human health”. This sentence can not be the conclusion of a work that do not present data about the effect of the cultivation on the environmental pollution.

Other considerations:

It is not clear the choice of the samples and the data presentation in Table 2 and 3, in particular the columns Laszka’, ‘Glen Ample’ and ‘Polka’ are referred to organic or conventional cultivation?

lines 84-87 The authors should describe accurately the method of preparation and identification of the phenolic compounds, including the sample preparation.

Line 100 After the first centrifugation (see the previous section)

As I said before, in the previous section there are not details about the sample preparation.

Line 252 “The variation in phenolic acid contents between cultivars is based on genetic factors. The raspberry cultivar ‘Polka’ contained 110.77 mg/100 g DW, whereas ‘Polana’ contained 125.65 mg/100 g DW 25. Among four raspberry cultivars, ‘Tulameen’ cv. was characterized by the lowest level of phenolic acids, with a value of 604.6 mg/100 g DW, and ‘Willamette’ had the highest level (1021.4 mg/100 g DW)”

The cultivars Polana and Tulameen are mentioned but are not between the samples examined in the paper. It is not clear

Author Response

Reviewer no. 4

Thank you very much for the review and for your recommendation to publish our manuscript in the “Antioxidant” journal.

Comment 1: “…The authors don’t give a hypothesis about the link between the antioxidant compounds content and the values of Bet v1 and profilin; in addition the data presented don’t show any correlation. In my opinion this work can not be published in the present form.…”

Authors’ response: According to Reviewer suggestion the preliminary hypothesis was added to proper section of presented manuscript. In Table 4 Authors presented the results of correlation analysis. The obtained data after statistically elaboration showed, that the correlation is weak (but significant in 2013) and much stronger (and still significant in 2014).

Comment 2: “…In the Conclusions the authors said : “Considering these results along with the other beneficial aspects of organic cultivation, less environmental pollution and production of more controlled food, that this second solution is much more beneficial for human health”. This sentence can not be the conclusion of a work that do not present data about the effect of the cultivation on the environmental pollution...…”

Authors’ response: Authors agree with Reviewer remark. Conclusions should be formatted strict on the basis on the experiment and the obtained results. In the presented experiment there were no any actions connected with environmental pollution. Authors give their apologise  and according to Reviewer suggestion changed wrong formulated conclusions.

Comment 3: “….It is not clear the choice of the samples and the data presentation in Table 2 and 3, in particular the columns ‘Laszka’, ‘Glen Ample’ and ‘Polka’ are referred to organic or conventional cultivation?..…”

Authors’ response: Authors want to clarified data presented in Table 1 and 2. The first two columns of the tables give values ​​for organic and conventional raspberries regardless of the cultivars. Organic raspberries = ‘Laszka’, ‘Glen Ample’ and ‘Polka’ fruits only from organic production. Conventional raspberries =  ‘Laszka’, ‘Glen Ample’ and ‘Polka’ from conventional production. In the next columns data presented are for individual cultivars (‘Laszka’ = fruits form organic and conventional production, but only ‘Laszka’ cv., ‘Polka’ = fruits form organic and conventional production, but only ‘Polka’ cv., ‘Glen Ample’ = fruits form organic and conventional production, but only ‘Glen Ample’ cv. Those data present all values ​​regardless of which system the cultivars came from.

Comment 4: “….lines 84-87 The authors should describe accurately the method of preparation and identification of the phenolic compounds, including the sample preparation...…”

Authors’ response: According to Reviewer suggestion  detailed description of polyphenols analysis was added into proper manuscript section.

Comment 5: “….Line 100 After the first centrifugation (see the previous section). As I said before, in the previous section there are not details about the sample preparation...”

Authors’ response: Authors want to apologise for lack of logic. Because a detailed description of polyphenol analysis was added in the previous section, the next sentence now sounds logically.

Comment 6: “….Line 252 “The variation in phenolic acid contents between cultivars is based on genetic factors. The raspberry cultivar ‘Polka’ contained 110.77 mg/100 g DW, whereas ‘Polana’ contained 125.65 mg/100 g DW 25. Among four raspberry cultivars, ‘Tulameen’ cv. was characterized by the lowest level of phenolic acids, with a value of 604.6 mg/100 g DW, and ‘Willamette’ had the highest level (1021.4 mg/100 g DW)”

The cultivars Polana and Tulameen are mentioned but are not between the samples examined in the paper. It is not clear…”

Authors’ response: Authors want to apologize for the unfortunate sentence. The ‘Tulameen’ and ‘Polana’ raspberry cultivars were the subject of the Pavlović experiment, not described experiment.

Pavlović, A.V.; Papetti, A.;  Zagorać, D.ÄŒ.D.; Gašić, U.M.; Mišić, D.M.; Tešić, Z.J.; Natić, M.M. Phenolics composition of leaf extracts of raspberry and blackberry cultivars grown in Serbia. Ind. Crop Prod. 2016, 87, 304–314.

Because other Reviver of presented manuscript pointed as well for that problem, sentence was corrected and write more accurately:

“…The variation in phenolic acid contents between cultivars is based on genetic factors. Experiment presented by Pavlović et al [35] among four raspberry cultivars, ‘Tulameen’ cv. was characterized by the lowest level of phenolic acids, with a value of 604.6 mg/100 g DW, and ‘Willamette’ had the highest level (1021.4 mg/100 g DW)….”

Round 2

Reviewer 3 Report

After reviewing the authors' response, the following points still requires adressing: 

Despite the preliminary nature of the study, I still think that all provided information has to be correctly addressed and interpreted. The statement that  ’In dark berry fruits, the anthocyanins could be responsible for causing allergic reactions’, does not reflect the conclusion of the study of Hjerno et al. and should be modified. This was not addressed by the authors.

The authors did not include the results of the ELISA assay as they mentioned they did.

The included fragments in the manuscript require English revision.

Author Response

Thank you very much for the review and for your positive recommendation to change our manuscript to be ready for publish in the “Antioxidant” journal.

Comment 1: “…Despite the preliminary nature of the study, I still think that all provided information has to be correctly addressed and interpreted. The statement that  ’In dark berry fruits, the anthocyanins could be responsible for causing allergic reactions’, does not reflect the conclusion of the study of Hjerno et al. and should be modified. This was not addressed by the authors..…”

Authors’ response: The authors want to apologize for the incorrect interpretation of the conclusions of Hjernø et al. (2006). As suggested by the Reviewer, the misinterpretation has been corrected in the manuscript text.

The interpretation was change from: “…’In dark berry fruits, the anthocyanins could be responsible for causing allergic reactions …” into: “…Strawberry allergen has an impact on the pathway for the synthesis of enzymes responsible for the synthesis of anthocyanins. Therefore, a lower allergen content directly affects a lower anthocyanin content. White strawberries (colour-less strawberry mutants) are known to be tolerated by individual affected by allergy, were found to be virtually free from the strawberry allergen….”

Comment 2: “…The authors did not include the results of the ELISA assay as they mentioned they did...…”

Authors’ response: According to the Reviewer's suggestion two tables with individual results of ELISA analysis were added to results section.

Comment 3: “…..The included fragments in the manuscript require English revision..…”

Authors’ response: Whole manuscript was checked by professional language company American Journal Experts (certificate in attachment)

Reviewer 4 Report

The authors have made the required changes. The paper can be accepted in the new form.

Author Response

Thank you very much for the review and for your recommendation to publish our manuscript in the “Antioxidant” journal.